# DIFFUSION-MODEL LAYERS MAY EXHIBIT DIFFUSIVE BEHAVIOR AT EACH STEP FOR NOISE ESTIMATION

## ABSTRACT

How do diffusion models process inputs at each step? The model transforms the input toward higher noise levels until it reaches pure noise prediction. We hypothesize that model layers exhibit diffusive behavior, which manifests as an internal, gradual diffusion process at each sampling step. Based on this insight, we introduce Depth-varying Diffusion (DvD), characterized by two key features: *1) Progressively stacking model layers across sampling steps (i.e., from T to 0) until it reaches the baseline depth.* As sampling progresses, the residual noise within the input becomes subtle, requiring more intense diffusion, and thus a deeper model, to map it into pure noise. *2) Enforcing supervision on both intermediate and final outputs.* Since stacked layers yield a cascade of "input $\rightarrow$ intermediate state $\rightarrow$ pure noise", we propose that the input should reach a specific intermediate state during the gradual diffusion process. We mathematically derive that in DvD, and correspondingly use the derived results for supervision. Experimental results demonstrate the effectiveness of these two features, showing improvements in generation quality while also reducing inference cost.

## 1    INTRODUCTION

Diffusion models Ho et al. (2020); Nichol & Dhariwal (2021); Song et al. (2022); Luo et al. (2022); Song et al. (2021) have emerged as a powerful generative framework in recent years, demonstrating remarkable performance in image synthesis Ho & Salimans (2022); Song et al. (2022) and various applications, including image editing Kawar et al. (2023); Yang et al. (2022a), speech Jeong et al. (2021) and video generation Ho et al. (2022); Singer et al. (2022). In essence, diffusion models generate images by starting from random noise and systematically denoising it through multiple sampling steps. At each sampling step, the diffusion model predicts the noise present in its input (which is a mixture of noise and image signal) and then removes this noise component. Repeating this process over multiple steps gradually transforms the initial noise into the generated image along a specific trajectory, either via a stochastic path method, SDE (*e.g.*, DDPM Ho et al. (2020)) or a deterministic path method, PF ODE (*e.g.*, DDIM Song et al. (2022) ). This paper focuses on the mechanism by which the diffusion model estimates the noise within its input at individual sampling steps (rather than the overall trajectory).

One key question we explore is how the model transforms the input into the predicted noise output at each step. We propose a hypothesis: model layers exhibit diffusive behavior, which manifests as an internal, gradual diffusion process at each sampling step, progressively moving the input toward pure noise prediction in each forward pass. Our hypothesis might seem counterintuitive, as it suggests that each denoising step actually uses internal diffusion for noise estimation. However, this does not alter the fundamental equations or training approaches in diffusion models. Instead, it provides new insights into how these models function step by step.

Based on this hypothesis, we further exploit two characteristics to enhance the noise estimation. First, during the entire sampling steps from $T \rightarrow 0$, the input noise ratio decreases from high to low, while the output noise is always expected to be pure, thereby requiring deeper layers to achieve greater diffusion rates. Second, since diffusion is a gradual process, the input should transition through a specific intermediate state before reaching the pure noise prediction. Obtaining this intermediate state ensures a more stable transition throughout the process.

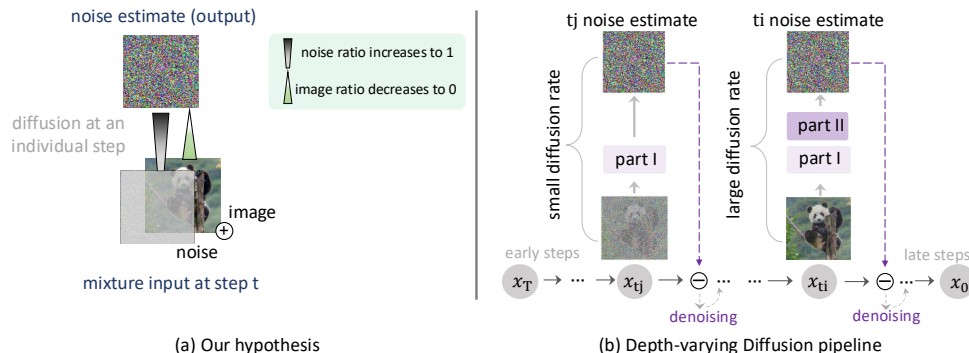

(a) Our hypothesis          (b) Depth-varying Diffusion pipeline

Figure 1: The illustration of our hypothesis and the proposed Depth-varying Diffusion (DvD). (a) We focus on each individual sampling step, within which the diffusion model is expected to transform the input (the mixture of image signal and random noise) into a pure noise output. We hypothesize that the model achieves this noise estimate through diffusion, gradually increasing the noise ratio to 1 and decreasing the image signal ratio to 0. (b) During the entire sampling steps from $T \rightarrow 0$, the input noise ratio gradually decreases while the output noise is expected to be pure. Therefore, the early steps and late steps require small and large diffusion rates, respectively. The proposed DvD uses shallow and deep model depth for early and late steps, catering to the nature that the required diffusion rates gradually increase along consecutive sampling steps. DvD maintains the total model size unchanged, reduces the training and inference costs, and improves generation quality.

These two points motivate us to propose a novel Depth-varying Diffusion (DvD), which progressively increases the model depth across the entire sampling steps, as shown in Fig. 1. Compared with the standard diffusion models (*e.g.*, DiT Peebles & Xie (2023)) that use constant model depth, our DvD model has a lower inference cost and maintains the same model size. This efficiency is achieved by initially using a relatively smaller model depth and progressively expanding to the baseline depth only during the final sampling steps. For implementation, we divide a holistic baseline model into several parts and progressively stack the involved parts. Without loss of generality, we take a two-part implementation for example (as in Fig. 1). DvD uses only the lower part (*e.g.*, Part I) for the early sampling steps and stacks the upper part (*e.g.*, Part I + Part II) for the late sampling steps. Empirically, we show our Depth-varying design not only improves the image generation quality, but also reduces the inference cost.

In addition to the Depth-varying structure, another feature of DvD is the supervision of intermediate states. As the stacking of layers increases, we enforce supervision to all involved parts, ensuring that each part is effectively optimized for its specific role. Specifically, in the relayed diffusion process from Part I $\rightarrow$ Part II, we consider the question: if Part I is capable of diffusing an early-step input into pure noise, how much diffusion should it accomplish given a late-step input, which is harder to diffuse? We address this question through a Taylor expansion of the diffusion process. The results show that Part I should increase the noise to a certain intermediate state. Therefore, during training at later sampling steps, we use this intermediate state as the supervision target for Part I, and leave the rest diffusion from intermediate to pure noise to be completed by the upper layers.

The simultaneous supervision strategy described above brings about another round of substantial improvement in generation quality. This improvement is especially encouraging because it demonstrates that the efforts of multiple parts could be cascaded and relayed (input $\rightarrow$ intermediate state $\rightarrow$ pure noise output), which indeed aligns with real-world diffusion processes. This observation supports our hypothesis that the model uses the diffusion to perform noise estimation.

Our main contributions are summarized as follows: First, we propose the hypothesis that diffusion model layers exhibit diffusive behavior, which manifests as an internal, gradual diffusion process to estimate the pure-noise output at each sampling step. Based on this hypothesis, we further exploit two characteristics of diffusion to enhance noise estimation. Second, we propose Depth-varying Diffusion (DvD) that progressively increases model depth across the entire sampling process. Third, we demonstrate that, in a single forward pass, the diffusion efforts of multiple model layers can be cascaded, allowing us to derive the intermediate states through the input's transition to the pure

noise prediction. Experimental results consistently validate the effectiveness of our proposed DvD, demonstrating substantial improvements in generation quality.

## 2 RELATED WORKS

**Diffusion Models.** Diffusion models Ho et al. (2020); Ho & Salimans (2022); Song et al. (2022); Rombach et al. (2022); Balaji et al. (2023); Podell et al. (2023); Nichol et al. (2022); Ramesh et al. (2022); Saharia et al. (2022); Gal et al. (2022); Nichol & Dhariwal (2021); Ho et al. (2021); Dhariwal & Nichol (2021) have rapidly gained prominence in generative modeling, notably without the need for adversarial training. This rise is particularly catalyzed by the foundational work on DDPM Ho et al. (2020), which introduces the concept of diffusion process. Many efforts Ho & Salimans (2022); Karras et al. (2022); Lu et al. (2022); Song et al. (2022); Salimans & Ho (2022); Song et al. (2023) have focused on accelerating the denoising sampling strategy. For example, DDIM Song et al. (2022) suggests that non-Markovian diffusion processes can achieve same training objective of DDPM Ho et al. (2020) and accelerate the sampling in generative processes. Furthermore, to facilitate the training of Diffusion Models on limited computational resources, LDM Rombach et al. (2022) proposes training diffusion processes in compressed latent space.

This paper proposes a hypothesis for diffusion models at individual sampling steps, *i.e.*, each sampling step actually involves a gradual diffusion process. Based on this insight, we exploit two key characteristics of this process and introduce Depth-varying Diffusion.

**Diffusion Transformers.** Transformers Dosovitskiy et al. (2021); Vaswani et al. (2023); Bao et al. (2022) have demonstrated remarkable scaling properties when model size is increased Kaplan et al. (2020); Li et al. (2021). Recently, several works Yang et al. (2022b); Bao et al. (2023); Jabri et al. (2023) have successfully integrated transformers with diffusion processes. Diffusion Transformers (DiT) Peebles & Xie (2023) have showcased the scalability of transformer layers, marking a significant advancement in the diffusion models. Recent works Gao et al. (2023; 2024); Zhu et al. (2024) aim to improve the training efficacy of DiT. MDT Gao et al. (2023; 2024) utilizes the encoder-decoder architecture of MAE He et al. (2021) and introduces masked latent embeddings to refine the relationships among semantic pixels within images. SD-DiT Zhu et al. (2024) applies self-supervised learning principles to diffusion models, enhancing their learning correlation.

This paper builds on DiT Peebles & Xie (2023) and adapts the model depth to accommodate varying diffusion rates. We observe that the early sampling steps (with higher noise ratios) require a smaller diffusion rate to achieve pure noise, while the late steps need a larger one. We divide the full model into several parts and increase the model depth across sampling steps (from $T \to 0$). This Depth-varying scheme improves the image generation quality and also reduces the inference cost.

**Varying Processing Strategy for Diffusion Models.** Many recent efforts also consider to adopt varying strategy for Diffusion model Chen et al. (2024); Liu et al. (2024); Feng et al. (2023); Park et al. (2024a). Some works introduce multiple expert diffusion models, each focusing on a specific time-step region. For example, ERNIE-ViLG 2.0 Feng et al. (2023) divides all time-steps into several blocks and employs multiple denoising experts at different stages. MEME Lee et al. (2024) assigns distinct architectures to different time-step intervals, balancing convolution and self-attention operations. Switch-DiT Park et al. (2024a) uses gating networks to process features at different time-steps. DTR Park et al. (2024b) selects different model channels to handle features, optimizing the diffusion process by routing tasks appropriately. Denoising Diffusion Step-aware Models (DDSM) Yang et al. (2024) propose using models of different sizes for different time-steps, with each model is trained across all time-steps to possess comprehensive learning. Although previous works process inputs separately according to their time-steps, they typically apply supervision only to the final output, enforcing alignment with pure noise.

In this paper, in addition to supervising the final output, we estimate the intermediate states for all involved parts during late steps and use these intermediate states as additional supervision targets. Since the lower parts can diffuse early-step input into pure noise, we employ mathematical derivations to infer the outputs of these parts in late steps. This simultaneous supervision strategy leads to another round of substantial improvement.

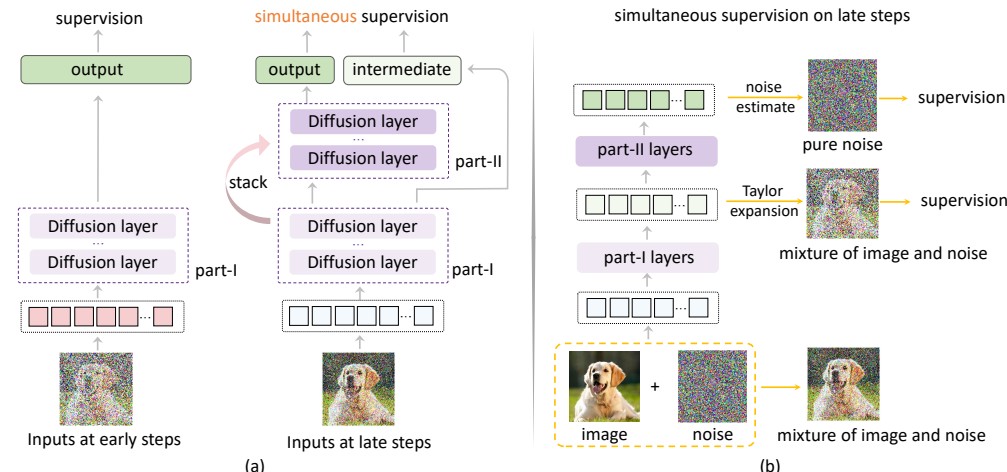

Figure 2: Overview of the Depth-varying Diffusion. Without loss of generality, we visualize the specific case of dividing the sampling steps into two intervals, *i.e.,* early and late steps. In **(a)**, Depth-varying Diffusion uses lower part (part I) for early steps and holistic parts (part I + part II) for late steps. The late steps share the early-step part I as the bottom and stack new part II on top. In **(b)**, we illustrate the simultaneous supervision strategy. We employ a Taylor expansion to estimate the intermediate state of Part I's output during late steps and use this estimate as a supervision target for Part I. The combination of supervision on Part I and the Part II constitutes the Simultaneous Supervision Strategy in late steps. More details are provided in Section 3.3. Depth-varying Diffusion can employ finer-grained division (*e.g.* 3 or 4 stages) and brings further improvement.

## 3 METHODS

### 3.1 OVERVIEW

**Preliminaries.** In diffusion models, the input at each sampling step is formulated as a mixture of the original image signal and random noise:

$$\mathbf{x}_t = \sqrt{\bar{\alpha}_t}\mathbf{x}_0 + \sqrt{1 - \bar{\alpha}_t}\boldsymbol{\epsilon}, \tag{1}$$

where $\boldsymbol{\epsilon}$ is random Gaussion noise drawn from a standard normal distribution, $\mathcal{N}(0, \mathbf{I})$, and $\mathbf{x}_0$ denotes the image signal. $\sqrt{\bar{\alpha}_t}$ and $\sqrt{1 - \bar{\alpha}_t}$ represent the ratios of image signal and noise, respectively.

During training, given the input $\mathbf{x}_t$ and the corresponding time-step $t$, the objective of the diffusion model is to predict the noise $\boldsymbol{\epsilon}$ from $\mathbf{x}_t$. Therefore, in each individual sampling, the model is expected to transform $\mathbf{x}_t$ into $\boldsymbol{\epsilon}$.

We view the transformation described above as being achieved through diffusion, and correspondingly define the increased ratio of noise as the diffusion rate $R_t$. The expectation of $R_t$ can be derived by: $R_t = 1 - \sqrt{1 - \bar{\alpha}_t}$. From the above definition, we derive the following Remarks.

**Remark-1**: Different sampling steps require different diffusion rates. More specifically, $R_t$ monotonically increases as the time-step $t$ progresses from $T \to 0$.

**Remark-2**: The input $\mathbf{x}_t$ should transition through a specific intermediate state before reaching the pure noise prediction $\boldsymbol{\epsilon}$.

Across the entire sampling steps, DvD progressively increases the model depth. The Depth-varying scheme is elaborated in Section 3.2. Section 3.3 further dwells into the gradual diffusion process within each individual forward and investigates its intermediate state(s) through a Tyler expansion. This results in a corresponding diffusion supervision manner, *i.e.,* simultaneously supervising the intermediate and ultimate output, and brings another round of generation quality improvement.

## 3.2 DEPTH-VARYING SCHEDULE

To progressively increase the model depth across time-step $t$, we divide the entire time-step range $[T, 0]$ into $N$ intervals and assign $i$-th interval a respective model depth $D_i$. We number the time-step intervals sequentially from 1 to $N$, with the first interval being the one closest to $T$. Dividing the time-step range is because it is usually larger than the model depth (*e.g.*, 1000 time-steps *vs* 12 layers in DiT-B Peebles & Xie (2023)).

For easy understanding, we start with a two-interval scenario as shown in Fig. 2 (*i.e.,,N = 2*) . We evenly divide the time-steps into two intervals, *i.e.*, early-step interval ( first interval ) $[T, T/2]$ and late-step interval (second interval) $[T/2, 0]$, assigning a model depth of $D_1 = L/2$ to the first interval and the holistic model depth of $D_2 = L$ to the second interval.

To achieve a more fine-grained partition (*i.e.*, $N > 2$), DvD recursively divides the early-step interval into two even sub-intervals. In this recursive procedure, DvD assigns a corresponding depth to each newly generated interval. We derive the two sub-intervals' depth with two criteria, *i.e.*, 1) the left-sub interval (closer to time-step $T$) has a smaller model depth than the right-sub interval (closer to time-step 0) , and 2) the average depth of the two sub-intervals remains equal to the depth of their parent interval, maintaining the inference cost unchanged. These two criteria are easily satisfied by:

$$D_{left} = D_{parent} - \delta, D_{right} = D_{parent} + \delta, \tag{2}$$

where the value of $\delta$ is a hyper-parameter optimized empirically. More details of the recursive strategy are provided in the ablation studies (Section 4.3).

**Noise estimation of DvD.** Given an input $\mathbf{x}_t$ , DvD locates its time-step $t$ in the pre-defined interval. Let us assume $t$ is within the $n$-th interval. DvD stacks the corresponding number of function parts (*i.e.*, $n$ parts) for diffusing $\mathbf{x}_t$ into the noise estimate $\mathbf{y}_t$, which is formulated as:

$$\mathbf{y}_t = F_n \circ F_{n-1} \circ \cdots \circ F_1(\mathbf{x}_t) \tag{3}$$

where $\circ$ denotes function composition, representing the sequential application of functions from $F_1$ to $F_n$. During the training, we use pure noise $\boldsymbol{\epsilon}$ as the supervision target for $\mathbf{y}_t$, according to the standard supervision:

$$\mathcal{L}^{\text{output}} = \mathbb{E}_{\boldsymbol{\epsilon}, \mathbf{y}_t} \left[ \|\boldsymbol{\epsilon} - \mathbf{y}_t\|^2 \right] \tag{4}$$

By progressively stacking a number of parts and applying pure-noise supervision at their final outputs, DvD explicitly trains each sub-part to handle the characteristic noise level of its corresponding interval. In practice, shallower parts quickly learn to diffuse highly noisy, early-step inputs toward pure noise, whereas deeper parts yield more intense diffusion for late-step inputs containing stronger image signals.

## 3.3 SIMULTANEOUS SUPERVISION STRATEGY

In addition to the standard supervision loss in Eqn. 4, DvD applies extra supervision on all the other involved parts, *i.e.*, from $F_1$ to $F_{n-1}$. A natural question thus arises: how to derive the supervision target of $F_k$ $(k < n)$? To simplify the following analysis, we use the two-part implementation as an example (shown in Fig. 2) and will later show that the analysis easily generalizes to multi-part scenario in appendix. We denote the lower / upper part as Part I / Part II, and use $F_1$ / $F_2$ to denote their underlying mapping function.

A clue to answer the above question is that: Part I is expected to diffuse an early-step input (closer to time-step $T$) into pure noise, according to its supervision target in Eqn. 4. With such diffusion ability and given a late-step input (closer to time-step 0), Part I should be able to increase certain ratio of noise, as well. We use $\mathbf{x}_{t_1}$ and $\mathbf{x}_{t_2}$ $(t_1 > t_2$ ) to denote the early-step and late-step input respectively and have:

$$\begin{aligned} \mathbf{x}_{t_1} &= \sqrt{\overline{\alpha}_{t_1}} \cdot \mathbf{x}_0 + \sqrt{1 - \overline{\alpha}_{t_1}} \cdot \boldsymbol{\epsilon} \\ \mathbf{x}_{t_2} &= \sqrt{\overline{\alpha}_{t_2}} \cdot \mathbf{x}_0 + \sqrt{1 - \overline{\alpha}_{t_2}} \cdot \boldsymbol{\epsilon} \end{aligned} \tag{5}$$

in which $\overline{\alpha}_{t_2} > \overline{\alpha}_{t_1}$ are two coefficients. Compared with the late-step input $\mathbf{x}_{t_2}$, the early-step input $\mathbf{x}_{t_1}$ contains less image signal and more noise component as $\mathbf{x}_{t_1}$ is closer to the initial sampling status ($t_1 > t_2$). Their qualitative relation can be expressed by:

$$\mathbf{x}_{t_1} = \mathbf{x}_{t_2} - k \cdot \mathbf{x}_0 + m \cdot \boldsymbol{\epsilon} \tag{6}$$

where

$$k = (\sqrt{\overline{\alpha}_{t_2}} - \sqrt{\overline{\alpha}_{t_1}}), m = (\sqrt{1 - \overline{\alpha}_{t_1}} - \sqrt{1 - \overline{\alpha}_{t_2}}) \tag{7}$$

Given that $F_1(\mathbf{x}_{t_1}) \to \boldsymbol{\epsilon}$, we try to figure out the value of $F_1(\mathbf{x}_{t_2})$. To this end, we perform Taylor expansion on $F_1(\mathbf{x}_{t_2})$, yielding:

$$\begin{aligned}
F_1(\mathbf{x}_{t_2}) =& F_1(\mathbf{x}_{t_1} - (m \cdot \boldsymbol{\epsilon} - k \cdot \mathbf{x}_0)) \\
=& F_1(\mathbf{x}_{t_1}) - \nabla F_1(\mathbf{x}_{t_1}) \cdot (m \cdot \boldsymbol{\epsilon} - k \cdot \mathbf{x}_0) \\
& - \frac{\nabla^2 F_1(\mathbf{x}_{t_1})}{2} \cdot (m \cdot \boldsymbol{\epsilon} - k \cdot \mathbf{x}_0)^2 + \cdots \\
=& k \cdot \nabla F_1(\mathbf{x}_{t_1}) \cdot \mathbf{x}_0 + (1 - \nabla F_1(\mathbf{x}_{t_1}) \cdot m)\boldsymbol{\epsilon} \\
& - \frac{\nabla^2 F_1(\mathbf{x}_{t_1})}{2} \cdot (m \cdot \boldsymbol{\epsilon} - k \cdot \mathbf{x}_0)^2 + \cdots
\end{aligned} \tag{8}$$

Based on the above formulation, we omit higher-order gradient terms and observe that the output of $F_1(\mathbf{x}_{t_2})$ can be approximated using $F_1(\mathbf{x}_{t_1})$ and its gradient $\nabla F_1(\mathbf{x}_{t_1})$. Empirical observations indicate that the mean value of $\nabla F_1(\mathbf{x}_{t_1})$ approaches 1 upon training convergence. Therefore, we simplify Eqn. 8 to:

$$F_1(\mathbf{x}_{t_2}) \to k \cdot \mathbf{x}_0 + (1 - m)\boldsymbol{\epsilon} \tag{9}$$

The above Eqn. 9 implies that if $F_1$ is capable of diffusing a early-step input $\mathbf{x}_{t_1}$ into pure noise, *i.e.*, $F_1(\mathbf{x}_{t_1}) \to \boldsymbol{\epsilon}$, it should diffuse an late-step input $\mathbf{x}_{t_2}$ into an intermediate state of $k \cdot \mathbf{x}_0 + (1 - m)\boldsymbol{\epsilon}$. Therefore, during training an later sampling step, we use $k \cdot \mathbf{x}_0 + (1 - m)\boldsymbol{\epsilon}$ as the intermediate supervision target for the output from Part I, which is formulated as:

$$\mathcal{L}^{\text{inter}} = \mathbb{E}_{\boldsymbol{\epsilon}, \mathbf{x}_{t_2}} \left[ \|k \cdot \mathbf{x}_0 + (1 - m)\boldsymbol{\epsilon} - F_1(\mathbf{x}_{t_2})\|^2 \right] \tag{10}$$

In overall, DvD combines supervision on the final output (Eqn. 4) and the intermediate state (Eqn. 10). The additional intermediate state supervision brings another substantial improvement in generation quality, as in Table 6.

**Discussion.** From $F_1(\mathbf{x}_{t_2}) \to k \cdot \mathbf{x}_0 + (1-m)\boldsymbol{\epsilon}$, it is observed that the output has larger noise than the input $\mathbf{x}_{t_2} = \sqrt{\overline{\alpha}_{t_2}} \cdot \mathbf{x}_0 + \sqrt{1 - \overline{\alpha}_{t_2}} \cdot \boldsymbol{\epsilon}$. This comparison indicates that Part I indeed increase the noise ratio for the early-step input but is still not able to diffuse it into pure noise (because $(1 - m)\boldsymbol{\epsilon} < \boldsymbol{\epsilon}$). The remaining diffusion from the intermediate state to the pure noise output is to be implemented by the following Part II. Such a relay-to-diffuse manner (*i.e.,* input $\to$ intermediate state $\to$ pure noise output) aligns with the gradual characteristic of real-world diffusion process.

## 4 EXPERIMENTS

### 4.1 SETUP

**Model architecture.** Following the DiT framework, we base our configurations on DiT-S, DiT-B, and DiT-XL. Our standard configuration includes a patch size of p=2. Additionally, we employ the standardized VAE from Stable Diffusion Rombach et al. (2022) for encoding and decoding image and latent tokens.

**Training details.** We perform all experiments on ImageNet-1K Russakovsky et al. (2015) at a resolution of 256×256 pixels. During training, we utilize the AdamW optimizer Loshchilov & Hutter (2019), setting the batch size to 256. The maximum time-step is 1000. All experiments are conducted on 8 × A100 GPUs.

**Evaluation.** During the evaluation phase, We assess the scaling performance using the Frechet Inception Distance (FID) Heusel et al. (2018), a key metric for evaluating image generative models.

Table 1: Comparison with DiT baseline across various model sizes on the 256×256 ImageNet dataset. DvD adopts DiT-S and DiT-B as baselines and compares the FID after 400K training steps.

| Method | Training Steps(k) | FID-50K↓ |
|---|---|---|
| Baseline#1 (DiT-S) Peebles & Xie (2023) | 400 | 68.40 |
| MDT-S Gao et al. (2023) | 400 | 53.46 |
| DvD-2P | 400 | **50.74** |
| DvD-3P | 400 | **50.43** |
| DvD-4P | 400 | **48.20** |
| Baseline#2 (DiT-B) Peebles & Xie (2023) | 400 | 43.47 |
| MDT-B Gao et al. (2023) | 400 | 34.33 |
| DvD-2P | 400 | **31.77** |
| DvD-3P | 400 | **31.56** |
| DvD-4P | 400 | **30.29** |

Table 2: Comparison with DiT baseline across various model sizes on the 256×256 ImageNet dataset. DvD uses the DiT-XL as the baseline.

| Method | Cost(Iter×BS) | FID-50K↓ |
|---|---|---|
| VQGAN Esser et al. (2021) | - | 15.78 |
| BigGAN-deep Brock et al. (2019) | - | 6.95 |
| I-DDPM Nichol & Dhariwal (2021) | - | 12.26 |
| Baseline#3 (DiT-XL) Peebles & Xie (2023) | 400k×256 | 19.47 |
| MDT-XL Gao et al. (2023) | 400k×256 | 16.42 |
| DvD-2P | 400k×256 | **16.09** |
| DvD-3P | 400k×256 | **16.01** |
| DvD-4P | 400k×256 | **15.74** |
| Baseline#3 (DiT-XL) Peebles & Xie (2023) | 7000k×256 | 9.62 |
| MaskDiT-XL Zheng et al. (2024) | 1300k×256 | 12.15 |
| MDT-XL Gao et al. (2023) | 1300k×256 | 9.60 |
| MDT-XL Gao et al. (2023) | 2500k×256 | 7.41 |
| MDT-XL Gao et al. (2023) | 3500k×256 | 6.46 |
| DvD-2P | 1300k×256 | **7.43** |
| DvD-2P | 2500k×256 | **5.58** |
| DvD-2P | 3500k×256 | **5.00** |

In line with established practices Peebles & Xie (2023); Gao et al. (2023), we compare our results to previous studies by reporting FID-50K scores. We utilizie the ADM's TensorFlow evaluation suite for measurement and also evaluate the Inception Score, sFID, and Precision/Recall.

## 4.2 MAIN RESULTS

DvD is evaluated on the ImageNet-1K Russakovsky et al. (2015) dataset. As shown in Table 1, we compare DvD with DiT-S and DiT-B. We employ three partitioning strategies for use in DvD, resulting in DvD-2P, DvD-3P, and DvD-4P, where time-steps are partitioned into 2, 3, and 4 intervals, respectively. The results in Table 1 lead to two key observations:

**1)** DvD consistently improves performance across all baselines. For example, under 400K training steps, DvD-2P improves DiT-S by 17.66 (68.40→50.74 ) and improves DiT-B by 11.7 (43.47→31.77). When using finer-grained partitions (*i.e.*, DvD-3P, DvD-4P), the performance of DvD continues to improve. For example, based on the DiT-S baseline, DvD-4P further improves the performance of DvD-2P from 50.74 to 48.20.

**2)** DvD demonstrates competitive performance. Notably, while the state-of-the-art method (MDT) leverages additional mask embeddings and incorporates a suite of best practices, DvD-4P surpasses MDT-B by 4.04, achieving an score of 30.29.

As shown in Table 2, We adopt DiT-XL as a strong baseline and compare DvD with state-of-the-art methods. The results further confirm the effectiveness. For example, after 400K training steps, DvD-2P improves DiT-XL by 3.38, achieving an FID of 16.09. DvD-4P further improves DvD-2P, improving the FID from 16.09 to 15.74, yielding improvement of 0.35. Moreover, DvD-2P (under

Table 3: DvD is compatible with classifier-free guidance and brings consistent improvements.

| Method | Cost (Iter×BS) | FID↓ | sFID↓ | IS↑ | Precision ↑ | Recall ↑ |
|---|---|---|---|---|---|---|
| DiT-XL-CFG | 7000k×256 | 2.27 | 4.60 | 278.2 | 0.83 | 0.57 |
| DvD-CFG | 4200k×256 | **2.06** | **4.52** | **283.0** | 0.83 | **0.60** |

Table 4: Comparison of DvD and DiT with DDIM. Both methods are trained for 400K steps.

| Method | Training steps | Sampling steps | FID ↓ |
|---|---|---|---|
| DiT-S (DDIM) | 400K | 100 | 69.90 |
| DvD-2P (DDIM) | 400K | 100 | 52.61 |
| DiT-S (DDIM) | 400K | 50 | 70.85 |
| DvD-2P (DDIM) | 400K | 50 | 54.00 |

2500K training steps) improves DiT-XL (under 7000K training steps) by 4.04 and achieves the score of 5.58. Compared with recent state-of-the-art method MDT-XL, DvD-2P surpasses MDT-XL and achieves 5.00 FID under 3500K steps. Table 3 validates DvD compatibility with Classifier-Free Guidance Ho & Salimans (2022), a well-established technique that improves diffusion models. DvD-CFG (trained for 4200K steps) surpasses DiT-XL-CFG (trained for 7000K steps) by 0.21, achieving a score of 2.06.

Table 5: Comparison of Depth-varying Diffusion with different varying strategies.

| Method | Time-steps | Layers | Flops(G) |
|---|---|---|---|
| DvD-2P[1] | [1000, 500] | 6 | 16.40 |
| DvD-2P[2] | [500, 0] | 12 | |
| DvD-3P[1] | [1000, 750] | 4 | |
| DvD-3P[2] | [750, 500] | 8 | 16.40 |
| DvD-3P[3] | [500, 0] | 12 | |
| DvD-4P[1] | [1000, 875] | 3 | |
| DvD-4P[2] | [875, 750] | 5 | 16.40 |
| DvD-4P[3] | [750, 500] | 8 | |
| DvD-4P[4] | [500, 0] | 12 | |

Table 6: Investigation on Depth-varying scheme and simultaneous supervision scheme.

| Method | Step(K) | FID ↓ |
|---|---|---|
| DiT-B (baseline) | 400 | 43.47 |
| DvD-2P w/o simu-loss | 400 | 35.63 |
| DvD-2P (full) | 400 | 31.77 |
| DvD-3P w/o simu-loss | 400 | 34.76 |
| DvD-3P (full) | 400 | 31.56 |
| DvD-4P w/o simu-loss | 400 | 34.62 |
| DvD-4P (full) | 400 | 30.29 |

## 4.3 ABLATION STUDIES

**Ablation on the initial two-interval partition scenario**. We investigate different partitioning strategies for the initial two-interval scenario. As shown in Fig. 3, we adopt DiT-B as baseline and select threshold timestep for dividing entire time-steps into two intervals. Additionally, we assign different model depths to the early-step interval to observe the performance variations. The results indicate that dividing the time-steps at $T/2$ and assigning half of the layers $L/2$ to the early-step interval achieves the best.

**Analysis on fine-grained Depth-varying scheme**. To achieve a more fine-grained partition, DvD recursively divides the first time-step interval into two even sub-intervals. For instance, to achieve three intervals, DvD divides $[T, T/2]$ into $[T, 3T/4]$ plus $[3T/4, T/2]$, and maintains the late interval $[T/2, 0]$ unchanged. According to the Eqn. 2, we design three partitioning schemes, $\delta$ is a hyper-parameter and is computed from the initial two-interval scenario as $(D_{right} - D_{left})/2 - 1$. We list some partitioning results of DvD in Table 5, based on DiT-B baseline. It shows that DvD consistently reduces the overall inference cost by 25%, decreasing Flops(G) from 21.90 to 16.40 across all settings compared to the baseline. While using more fine-grained partition maintains the same efficiency, experimental results in Table 6 show that it slightly improves the generation quality.

In addition to the empirical improvements on the testing data, Fig. 4 further reveals that DvD enables the deep model to better fit the training data. Compared to the baseline model DiT-B, DvD exhibits lower prediction error across all time-steps.

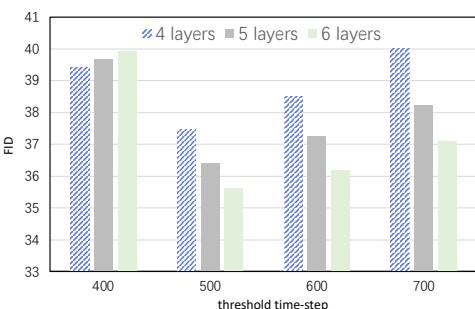
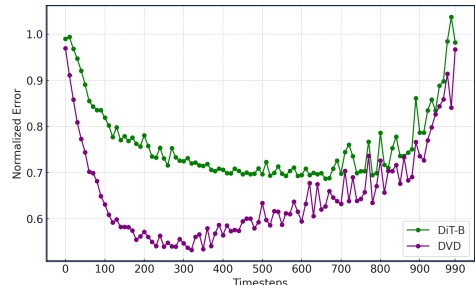

Figure 3: Comparison of different initial varying strategies for two-intervals.

Figure 4: The normalized prediction error between DvD and DiT on training data.

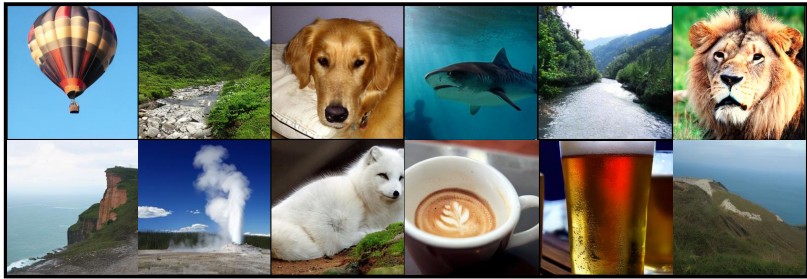

Figure 5: Images generated by DvD based on DiT-XL.

**Comparison of DvD and baseline model with DDIM.** Since both DDPM and DDIM generate images by starting from random noise and iteratively denoising through multiple sampling steps, at each step predicting the noise in their input and then removing it, we also evaluate the performance of DvD and DiT-S using the DDIM sampling method. As shown in Table 4, when applying DDIM, DvD improves DiT-S, improving its score from 69.9 to 52.61 with 100 sampling steps. Even with fewer sampling steps (*i.e.*, 50 steps), DvD continues to enhance DiT-S, improving the FID from 70.85 to 54. These results demonstrate that DvD is adaptable to different sampling methods and consistently yields performance improvements.

**Evaluation on important configurations.** As shown in Table 6, Depth-varying scheme consistently improves performance. For example, compared to baseline DiT-B, the results of DvD-2P, DvD-3P, and DvD-4P demonstrate improvements of +7.84, +8.71, and +8.85. Moreover, simultaneous supervision also enhances performance. For example, incorporating simultaneous supervision into DvD-2P, DvD-3P, and DvD-4P leads to performance gains of +3.86, +3.20, and +4.33, respectively.

**Visualization.** We visualize images generated by the proposed DvD-XL to intuitively show its generation quality in Fig. 5.

## 5 CONCLUSION

This paper focuses on the mechanism by which the diffusion model estimates the noise within its input at individual step and introduces a hypothesis, *i.e.*, diffusion model layers exhibit diffusive behavior for noise prediction. Guided by this hypothesis, we exploit two characteristics: 1) different sampling steps require different diffusion rates, and 2) the input should reach a specific intermediate state before reaching the pure noise prediction. Based on these insights, we propose Depth-varying Diffusion (DvD), a method that progressively increases its model depth across sampling steps in alignment with the growing diffusion rates and investigate the intermediate states of the input as it transitions to pure noise using a Taylor expansion. During training, we enforce simultaneous supervision on both intermediate states and final outputs. Experimental results consistently validate the effectiveness of our proposed DvD, demonstrating substantial improvements in generation quality.

## ETHICS STATEMENT

This paper improves both image generation quality and inference speed, making it applicable to more versatile image generation scenarios. This paper utilizes open dataset to train and evaluate the performance. We aim to further enhance the generalization of this method and reduce potential problems.

## REPRODUCIBILITY STATEMENT

The DvD is reproducible. In the main text, we describe the utilized datasets in DvD, *i.e.,* ImageNet. We provide the details about the experimental implementation, the proof of the proposed remark and the analysis of some hyper parameters in appendix.

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

# A    APPENDIX

In the appendix, we introduce more details that are not described in the main text. In SectionA.1, we provide more details of the Depth-varying strategy with additional baselines. In SectionA.2, we compare the convergence speed between DvD and DiT. In SectionA.3, we provide further analysis of the simultaneous supervision strategy. In SectionA.4, we illustrate more visualizations. In SectionA.5, we introduce the use of Large Language Models.

## A.1    MORE DETAILS OF DEPTH ALLOCATION STRATEGY

**Depth-varying strategy based on DiT-S.** As shown in Table A1, we use DiT-S as the baseline and list three partitioning strategies: DvD-2P, DvD-3P, and DvD-4P. We calculate the sub-Flops of the model at each time-step interval and report their average across all time-steps as the overall Flops. Compared to DiT-S, DvD reduces the Flops from 5.50 to 4.10 (a 25% reduction), while the Flops remain identical across all partitioning strategies.

Table A1: Comparison of Depth-varying Diffusion with different time-step partitioning strategies based on DiT-S.

| Method | Time-steps | Layers | sub-Flops(G) | Flops(G) |
|---|---|---|---|---|
| DiT-S | [1000, 0] | 12 | 5.50 | 5.50 |
| DvD-2P[1] | [1000, 500] | 6 | 2.75 | 4.10 |
| DvD-2P[2] | [500, 0] | 12 | 5.50 | |
| DvD-3P[1] | [1000, 750] | 4 | 1.83 | |
| DvD-3P[2] | [750, 500] | 8 | 3.66 | 4.10 |
| DvD-3P[3] | [500, 0] | 12 | 5.50 | |
| DvD-4P[1] | [1000, 875] | 3 | 1.38 | |
| DvD-4P[2] | [875, 750] | 5 | 2.29 | 4.10 |
| DvD-4P[3] | [750, 500] | 8 | 3.66 | |
| DvD-4P[4] | [500, 0] | 12 | 5.50 | |

**Depth-varying strategy based on DiT-XL.** We present the partitioning results of DvD based on DiT-XL in Table A2. The time-steps division follows the same strategy as applied to DiT-B / S , and the layer assignment for each time-step interval is derived from Eqn.2 in the main text. Compared to DiT-XL, DvD reduces the Flops from 114.78 to 86.10, achieving a reduction of 28.68, and the Flops remain identical across all partitioning strategies.

Table A2: Comparison of Depth-varying Diffusion with different time-step partitioning strategies based on DiT-XL.

| Method | Time-steps | Layers | sub-Flops(G) | Flops(G) |
|---|---|---|---|---|
| DiT-XL | [1000, 0] | 12 | 114.78 | 114.78 |
| DvD-2P[1] | [1000, 500] | 6 | 57.39 | 86.10 |
| DvD-2P[2] | [500, 0] | 12 | 114.78 | |
| DvD-3P[1] | [1000, 750] | 4 | 32.80 | |
| DvD-3P[2] | [750, 500] | 8 | 81.99 | 86.10 |
| DvD-3P[3] | [500, 0] | 12 | 114.78 | |
| DvD-4P[1] | [1000, 875] | 3 | 12.31 | |
| DvD-4P[2] | [875, 750] | 5 | 53.30 | 86.10 |
| DvD-4P[3] | [750, 500] | 8 | 81.99 | |
| DvD-4P[4] | [500, 0] | 12 | 114.78 | |

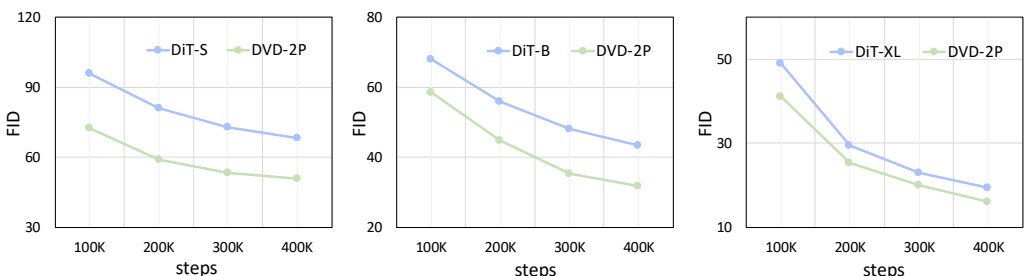

Figure A1: Comparison of training convergence speed between DiT and DvD.

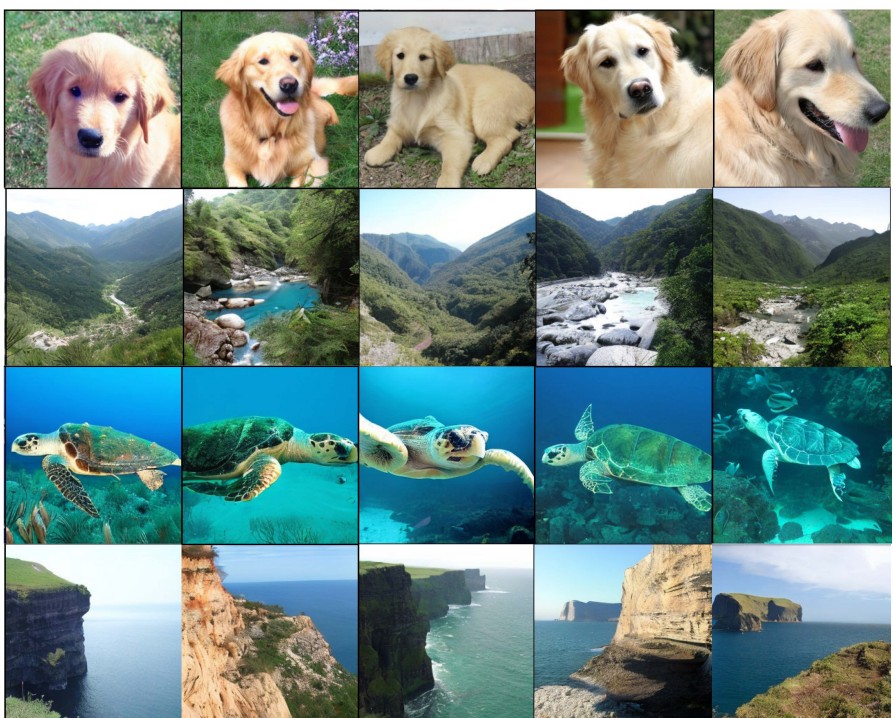

Figure A2: More Images generated by DvD based on DiT-XL.

## A.2 CONVERGENCE SPEED.

We compare the convergence speed from 0 to 400K training steps between DvD and baselines. The batch size is set to 256 for fair comparisons, and the maximum training step is 400K. As shown in Fig A1, DvD consistently improves training convergence across all DiT-S, DiT-B, and DiT-XL baselines. For example, based on the DiT-B, DvD-2P achieves a FID of 45.1 at 200K training steps, nearly twice as fast compared to baseline.

## A.3 SIMULTANEOUS SUPERVISION STRATEGY

In Section.3.3 of the main text, we use the two-part implementation as an example to derive intermediate states. For a more fine-grained partition scenario in which the total time-steps are divided into $N$ intervals, we denote the inputs at the $n$-th and $i$-th time-steps as $\mathbf{x}_{t_n}$ and $\mathbf{x}_{t_i}$ (with $t_n < t_i$), respectively, and have:

$$
\begin{aligned}
\mathbf{x}_{t_i} &= \sqrt{\overline{\alpha}_{t_i}} \cdot \mathbf{x}_0 + \sqrt{1 - \overline{\alpha}_{t_i}} \cdot \boldsymbol{\epsilon} \\
\mathbf{x}_{t_n} &= \sqrt{\overline{\alpha}_{t_n}} \cdot \mathbf{x}_0 + \sqrt{1 - \overline{\alpha}_{t_n}} \cdot \boldsymbol{\epsilon}
\end{aligned}
\tag{11}
$$

in which $\overline{\alpha}_{t_n} > \overline{\alpha}_{t_i}$ are two coefficients. Compared to the input $\mathbf{x}_{t_n}$, the input $\mathbf{x}_{t_i}$ contains less image signal and more noise since $\mathbf{x}_{t_i}$ is closer to the initial sampling state ($t_i > t_n$). Their qualitative relationship can be expressed as:

$$\mathbf{x}_{t_i} = \mathbf{x}_{t_n} - k \cdot \mathbf{x}_0 + m \cdot \boldsymbol{\epsilon} \tag{12}$$

where

$$k = (\sqrt{\overline{\alpha}_{t_n}} - \sqrt{\overline{\alpha}_{t_i}}), m = (\sqrt{1 - \overline{\alpha}_{t_i}} - \sqrt{1 - \overline{\alpha}_{t_n}}) \tag{13}$$

Given that $F_i \circ \cdots \circ F_1(\mathbf{x}_{t_i}) \to \boldsymbol{\epsilon}$, we aim to determine the value of $F_i \circ \cdots \circ F_1(\mathbf{x}_{t_n})$. To achieve this, we apply a Taylor expansion to $F_i \circ \cdots \circ F_1(\mathbf{x}_{t_n})$, resulting in:

$$
\begin{aligned}
&F_i \circ \cdots \circ F_1(\mathbf{x}_{t_n}) \\
&= F_i \circ \cdots \circ F_1(\mathbf{x}_{t_i} - (m \cdot \boldsymbol{\epsilon} - k \cdot \mathbf{x}_0)) \\
&= F_i \circ \cdots \circ F_1(\mathbf{x}_{t_i}) - \nabla F_i \circ \cdots \circ F_1(\mathbf{x}_{t_i}) \cdot (m \cdot \boldsymbol{\epsilon} - k \cdot \mathbf{x}_0) \\
&\quad - \frac{\nabla^2 F_i \circ \cdots \circ F_1(\mathbf{x}_{t_i})}{2} \cdot (m \cdot \boldsymbol{\epsilon} - k \cdot \mathbf{x}_0)^2 + \cdots \\
&= k \cdot \nabla F_i \circ \cdots \circ F_1(\mathbf{x}_{t_i}) \cdot \mathbf{x}_0 + (1 - \nabla F_i \circ \cdots \circ F_1(\mathbf{x}_{t_i}) \cdot m)\boldsymbol{\epsilon} \\
&\quad - \frac{\nabla^2 F_i \circ \cdots \circ F_1(\mathbf{x}_{t_i})}{2} \cdot (m \cdot \boldsymbol{\epsilon} - k \cdot \mathbf{x}_0)^2 + \cdots
\end{aligned}
\tag{14}
$$

We omit higher-order gradient terms and approximate the output of $F_i \circ \cdots \circ F_1(\mathbf{x}_{t_n})$ using $F_i \circ \cdots \circ F_1(\mathbf{x}_{t_i})$ and its gradient $\nabla F_i \circ \cdots \circ F_1(\mathbf{x}_{t_i})$. Therefore, we simplify Eqn. 14 to:

$$F_i \circ \cdots \circ F_1(\mathbf{x}_{t_n}) \to k \cdot \mathbf{x}_0 + (1 - m)\boldsymbol{\epsilon} \tag{15}$$

Finally, we obtain the $i$-th intermediate state of $\mathbf{x}_{t_n}$ in the multi-parts scenario.

## A.4 MORE VISUALIZATION RESULTS

We illustrate more images (uncurated) generated by Depth-varying-XL in Figure A2. The class labels are "golden retriever" (207), "valley"(979), "loggerhead turtle" (33) and "cliff drop-off" (972). The image resolution is 256×256.

## A.5 USE OF LARGE LANGUAGE MODELS

We use the Large Language Models to refine and polish the wording and grammar of the paper.

