# OpenReview forum: "Diffusion-Model Layers May Exhibit Diffusive Behavior at Each Step for Noise Estimation"
_ICLR.cc/2026/Conference — ICLR 2026 Conference Withdrawn Submission_

### Official Review · Reviewer_jkWc · 2025-10-30

**Soundness:** 1
**Presentation:** 2
**Contribution:** 1
**Rating:** 2
**Confidence:** 2

**Summary:**

This paper suggests depth-varying diffusion model along with time step t. The authors progressively stacked layers across sampling steps and applying simultaneous supervision. They derive the necessity of simultaneous supervision using Taylor expansion on diffusion evaluation.  With such suggested items, the authors achieved better results than baselines by a margin ~3-5 FID gap within same training cost.

**Strengths:**

- The authors catch some intuitions : noise level is different at each time step. Then, they fully realized such intuition by stacking additional model layers.

- They earned better FIDs against baseline algorithm within same training cost.

**Weaknesses:**

- limited novelty : the idea of DvD is simple and is realized with depth-varying architectures along time step. This may increase the computation time and memory consumption. Based on reported tables, there are no significant gains even with such trade-offs. I'm not sure of expansion capability of this work. The proposed method develops an algorithm with additional overloads.

- reasoning on their intuition is weak. The solution to address different diffusive rate is stacking additional layers which look initial brute-force trials. Especially, there is no reasonable evidence that time-step aware diffusion model is inferior to depth-varying diffusion model. The method to increase the number of layers could be well replaced with current time-step aware diffusion.

- datasets are too limited : algorithm was validated only on imagenet1k. Emprical evidences are weak.

- qualitative demonstrations are not satisfactory. It's not enough to show difference from baselines.

**Questions:**

1) In my knowledge, diffusion model is not a simple denoiser. It may become an attractor-generator in terms of repetitive evaluation on given noises like dynamic system (autonomous ODE system, Hopfield model):
" In search of dispersed memories: Generative diffusion models are associative memory networks," arXiv 23

Thus, if we use different depth or different weight network per each different time step, such attracted or associative property would be broken. Thus, DvD approach may harm in performances within equivalent diffusion setup such as equal number of trainable parameters.

I believe DvD approach use more layers and parameters than baselines. Please correct me if I'm wrong.

2) why does DvD-4p show best performances?

3) In table 5, the authors reported Flops analysis with different varying strategies. I see they tried to unify Flops across different partitioning scheme. Then, how is this changed for inference time?

---

### Official Review · Reviewer_CPtZ · 2025-10-31

**Soundness:** 2
**Presentation:** 1
**Contribution:** 2
**Rating:** 4
**Confidence:** 3

**Summary:**

The paper proposes Depth-varying Diffusion (DvD) for DiT-style diffusion models. The central hypothesis is that “diffusion model layers exhibit diffusive behavior,” i.e., within a single forward pass at step \(t\), deeper layers supposedly gradually transform the input toward a pure-noise prediction, an internal diffusion process. Building on this, DvD (i) uses fewer layers at early, high-noise steps and more layers at later, low-noise steps by stacking shared parts across time, and (ii) adds simultaneous supervision by training lower parts at late steps to match an intermediate target derived from a first-order Taylor approximation (their Eq. (8)–(10). Experiments on ImageNet-256 with DiT-S/B/XL report lower FID and about 25% fewer inference FLOPs than constant-depth baselines; ablations suggest the auxiliary loss helps.

**Strengths:**

- Practical scheduling: time-conditioned depth (shallow early, deep late) is simple, keeps parameters fixed, reduces average inference cost, and remains compatible with DDIM/CFG. The pipeline figures clarify the intended scheduling across timestep intervals.
- Auxiliary supervision plausibly helps optimization: the simultaneous supervision uses an intermediate target for lower parts at late steps, aligning with common deep-supervision practices.
- Reported gains on DiT baselines: Tables 1-2 show FID improvements with roughly 25% inference FLOP reductions.

**Weaknesses:**

- Unvalidated core claim: the statement that layers exhibit diffusive behavior (an internal, gradual diffusion within a single forward pass) is never tested. There are no probes such as feature or noise-ratio tracking across depth, or representation analyses - the claim is repeated across abstract, introduction, and methods and never verified.
- The experimental comparison is restricted to constant-depth DiT baselines. No results are shown against other methods that adjust model capacity or computation over diffusion timesteps, limiting clarity on the relative contribution of DvD within this class of techniques.
- Heuristic Taylor derivation: the intermediate target relies on assuming $\mathbb{E}[\nabla F_1(x_{t_1})]\approx 1$ and discarding higher-order terms (Eq. 8-10) without empirical verification (no gradient statistics, sensitivity, or error analysis).
- Presentation quality: the paper is not polished, repeated phrasing, informal mathematical definitions/remarks or undefined terms (e.g., diffusion layer), and typos (Gaussion, Tyler expansion) reduce clarity and rigor.

**Questions:**

1) Hypothesis validation: can you provide empirical probes confirming internal diffusion across layers at a fixed step? Currently the core hypothesis is untested.

2) Taylor-target sensitivity: please report statistics for $\nabla F_1(x)$ (mean and variance over training) and an ablation on the $\approx 1$ assumption and on truncation of higher-order terms in Eq. (8)–(10).

3) Could the authors provide comparisons to other approaches that adapt model depth or compute as a function of the timestep, to better situate DvD within this broader class of methods?
Even a compute-matched comparison to one such variant would help clarify the degree of improvement specific to DvD’s formulation.

---

### Official Review · Reviewer_xyc3 · 2025-11-01

**Soundness:** 3
**Presentation:** 3
**Contribution:** 4
**Rating:** 4
**Confidence:** 4

**Summary:**

This submission presents a hypothesis that layers in diffusion models perform diffusion by each layer, not necessary by the whole network.
This insight leads to propose  Depth-varying Diffusion (DvD) model, which 1)  uses the only earlier part of the network in the earlier steps of the reverse process since the noise-intensive images are easier to denoise, and gradually adding stacked layers in the later steps.
DvD is further enhanced with intermediate-supervision strategy that inject training denoising targets in even after later-step layers are stacked, which is achievable using Taylor-expansion-based conversion of the training targets.
Experiments are conducted using 256×256 ImageNet, and DvD is shown to improve FID scores DiT-based models without increasing model parameters and with reduced inference-time computation.

**Strengths:**

- The proposed method is practical by improving the generation results without increasing model parameters and with reduced inference-time computation.
- The insight behind the design of the method is non-traivial and interesting, because it is not clear that the parts and the whole can have similar behaviors in complex systems such as neural network.

**Weaknesses:**

- Ablation study of the method is not conducted. Especially I suspect that 1) earlier (Part I) layers are dedicated for denoising in the earlier steps, and later layers are dedicated for later steps and the stacking might not be necessary.
- Quantitative results in CFG experiments show only marginal performance gains. Is there any possible reasons?
- Experiments are solely conducted on ImageNet-1k and the method's generalizability in other domains is unclear. Reliability of the results would be strengthen by using different datasets, even smaller-scale ones.

**Questions:**

-  Is the timestep embedding used in DvD? Showing the model which step it is at explicitly may change the behavior.

---

### Official Review · Reviewer_Yvmy · 2025-11-01

**Soundness:** 2
**Presentation:** 1
**Contribution:** 1
**Rating:** 2
**Confidence:** 3

**Summary:**

This paper introduces Depth-varying Diffusion (DvD), a framework that adaptively adjusts the network depth across diffusion timesteps.
The key idea is that within each denoising step, the model behaves “diffusively,” so earlier, noisier steps can be handled by shallower subnetworks, while later, cleaner steps benefit from deeper ones. DvD further adds intermediate supervision via a Taylor-based approximation. Experiments on ImageNet show that DvD achieves better FID and up to ~25% lower inference cost compared to standard DiT models.

**Strengths:**

- **Consistent quantitative improvements.** Across model scales, DvD achieves lower FID-50K than DiT baselines while being compatible with standard samplers.
- **Extensive experimental comparisons** The method is evaluated on multiple DiT backbones (S/B/XL) and compared against various baselines, demonstrating consistent improvements across architectures and sampling settings.

**Weaknesses:**

- **Figure 1 is difficult to interpret.**
  Although Fig. 1(a) aims to illustrate the diffusion assumption and (b) the depth-varying pipeline, the figure lacks symbol definitions, data-flow arrows, and an explanation of which part of the assumption is being visualized. The caption alone is insufficient for independent understanding.

- **Weak motivation for noise-dependent depth.**
  The intuition that “shallower models suffice at high noise, deeper models at low noise” is plausible but not theoretically or empirically justified.

- **Limited applicability.**
  The method cannot be directly applied to v-prediction or latent diffusion models due to VAE reconstruction errors, which restricts its practical utility.

- **Ambiguity in the “cost-invariance” statement.**
  The claim that “the average depth of the two sub-intervals equals their parent, maintaining inference cost unchanged” is confusing—if both sub-intervals are executed sequentially, total computation may effectively double. This requires clarification.

- **Unclear boundaries between Part I and Part II.**
  The paper should specify the exact layer ranges or modules corresponding to each part in the experimental setup.

- **Insufficient analysis of “diffusive behavior.”**
  The paper assumes each block behaves as a local diffusion operator but provides no theoretical reasoning or comparison showing why this behavior is beneficial relative to existing architectures.

- **Missing ablation on intermediate-state supervision.**
  Although this is a key design component, no experiment isolates its contribution or verifies its necessity.

**Questions:**

See weakness

---

### Note · Authors · 2025-11-12

I have read and agree with the venue's withdrawal policy on behalf of myself and my co-authors.